# Prediction of Fatigue Crack Initiation of 7075 Aluminum Alloy by Crystal Plasticity Simulation

**DOI:** 10.3390/ma16041595

**Published:** 2023-02-14

**Authors:** Takayuki Shiraiwa, Fabien Briffod, Manabu Enoki

**Affiliations:** Department of Materials Engineering, School of Engineering, The University of Tokyo, 7-3-1 Hongo, Bunkyo-ku, Tokyo 113-8656, Japan

**Keywords:** aluminum alloy, fatigue, crystal plasticity, finite element method, crack initiation

## Abstract

The 7075 aluminum alloy is a promising material for the aerospace industry due to its combination of light weight and high strength. This study proposed a method for predicting fatigue crack initiation of the 7075 aluminum alloy by crystal plasticity finite element analysis considering microstructures. In order to accurately predict the total fatigue life, it is necessary to calculate the number of cycles for fatigue crack initiation, small crack growth, and long crack growth. The long crack growth life can be estimated by the Paris law, but fatigue crack initiation and small crack growth are sensitive to the microstructures and have been difficult to predict. In this work, the microstructure of 7075 aluminum alloy was reconstructed based on experimental observations in the literature and crystal plasticity simulations were performed to calculate the elasto-plastic deformation behavior in the reconstructed polycrystalline model under cyclic deformation. The calculated local plastic strain was introduced into the crack initiation criterion (Tanaka and Mura, 1981) to predict fatigue crack initiation life. The predicted crack initiation life and crack morphology were in good agreement with the experimental results, indicating that the proposed method is effective in predicting fatigue crack initiation in aluminum alloys. From the obtained results, future issues regarding the prediction of fatigue crack initiation were discussed.

## 1. Introduction

The 7000 (Al-Zn-Mg) and 2000 (Al-Cu-Mg) series aluminum alloys are widely used as structural materials in aircraft and space applications. The stable precipitate phases in Al-Zn-Mg alloys are MgZn_2_ (η phase) and Mg_3_Zn_3_Al_2_ (T phase) [1]. The metastable phases (η’, T’ phases) that appear during the transition from the spherical GP zone to the stable phase are fine and produce substantial lattice strains, resulting in excellent precipitation strengthening. The 7075 alloy (Al-5.6Zn-2.5Mg-1.6Cu-0.23Cr in wt%) is a representative 7000 series alloys, further enhanced with the addition of Cu and Cr, and is commonly utilized in a fully precipitated state by T6 heat treatment. While the 7075 alloys have the highest tensile strength among aluminum alloys, they are inferior in fatigue strength and corrosion resistance, making them difficult to use in parts subjected to tensile fatigue loading. Their application is limited to parts subjected to compressive fatigue loading, such as the upper wing surface. Since aircraft are subjected to cyclic loading due to takeoffs, landings, and changes in atmospheric pressure, further improvement in fatigue performance is required.

The fatigue failure of metallic materials occurs when the material is subjected to repeated loads, resulting in irreversible cyclic plastic deformation, crack initiation, the small crack growth, the long crack growth, and final failure. The behavior of long crack growth can be predicted based on linear fracture mechanics. However, the prediction of fatigue crack initiation and the microstructurally small fatigue crack growth remains challenging. Since the work by Forthyth [2], many observations have been made regarding fatigue crack initiation in aluminum alloys. Under the positive stress ratio (tensile fatigue loading), fatigue crack initiations from inclusions [3] and voids [4] have been reported in 7000 series alloys, but the effect of the type and size of the inclusions on crack initiation is not clear. On the other hand, it has been observed that under conditions of stress ratios below zero, as used in practical applications, fatigue cracks are generated by the formation of transgranular facets at the surface of the material [5,6]. In order to predict such fatigue crack initiation within grains, it is necessary to analyze the cyclic plastic deformation behavior of polycrystalline materials considering the grain morphologies and crystallographic orientations.

The crystal plasticity finite element method (CPFEM) is a suitable numerical method for such purposes and has been used to predict fatigue properties in steels [7], titanium alloys [8], and aluminum alloys [9,10,11]. In these analyses, fatigue crack initiation life is often predicted by quantifying the driving force for fatigue crack initiation as a fatigue indicator parameter (FIP) [12], which is defined by the Tanaka-Mura model based on dislocation theory [13] and/or the Fatemi-Socie model based on critical plane theory [14]. However, methods for quantifying the FIP from the stress-strain field computed by CPFEM are still not well established. Previous papers on fatigue prediction of aluminum alloys have several drawbacks, such as the inability to predict crack shape considering the plastic anisotropy of each grain and the inability to relate the slip band length to grain morphology in the Tanaka-Mura model. Our research group has proposed a method to solve these problems by introducing potential crack paths parallel to the slip bands, and applied the method to pure iron [15], titanium alloys [16], and magnesium alloys [17].

The purpose of this paper is to investigate the applicability of fatigue crack initiation prediction by the crystal plasticity simulation to aluminum alloys. Among the 7000 series alloys, the 7075 alloy was selected as the target of this study because of the abundance of experimental fatigue data. The fatigue prediction results by CPFEM and potential crack paths are compared with the experimental data in the literature. The physical interpretation of the fatigue crack initiation criterion will be also discussed.

## 2. Materials and Methods

### 2.1. Materials

The material investigated in this study is 7075 aluminum alloy. Numerical models in this paper were constructed and calibrated with reference to experimental data of the 7075 aluminum alloy from the literature [5,18]. In the literature, a rolled 7075 aluminum alloy was subjected to T6 heat treatment, i.e., solution heat treatment at 480 °C for 1 h followed by aging at 120 °C for 24 h. The grains were elongated in the rolling direction, with average dimensions of 600 µm in the rolling direction (RD) and 108 µm in the transverse direction (TD). The tensile strength was 623 MPa, and elongation was 16%. The inverse polar figure map obtained by electron back-scatter diffraction (EBSD) experiments [18] are shown in Figure 1a.

### 2.2. Generation of Synthetic Microstructures

As mentioned in the previous section, the material has an anisotropic grain morphology. To reproduce such grain morphology, a polycrystalline aggregate model was constructed by anisotropic weighted Voronoi tessellation [15]. In this tessellation method, ellipses are sampled from assumed statistical distributions of grain shapes and positioned in model space by a relaxed random sequential addition (RSA) algorithm [19]. Then, an anisotropic tessellation is performed using the center positions of the ellipses as seeds, and finally the crystal orientation is assigned. In this study, the statistical distributions of grain shape parameters were assumed based on the experimental data. The major axis length was assumed to be normally distributed with a mean of 600 μm and a standard deviation of 50 μm, the aspect ratio of the major axis to the minor axis was normally distributed with a mean of 0.2 and a standard deviation of 0.05, and the angle between the major axis and RD was normally distributed with a mean of 0° and a standard deviation of 5°. The crystal orientation was assumed uniformly random for simplicity. A synthetic polycrystalline microstructure generated using these statistical distributions is shown in Figure 1b. It is noted that the polycrystalline structure is periodic in the x- and y-directions (RD and TD). Considering the computational cost, the model size and mesh size were set to 1500 μm × 3000 μm and 5 μm, respectively. The number of elements was 180,000. Three-dimensional eight-node hexahedral elements (C3D8) were used for meshing.

### 2.3. Constitutive Law and Parameter Calibration

To analyze the elasto-plastic deformation behavior of the polycrystalline model under cyclic loading, a constitutive law based on crystal plasticity theory was used in the finite element analysis. The total deformation gradient **F** can be decomposed into an elastic part **F**_e_ and a plastic part **F**_p_:(1)$$F=F_{e}F_{p}$$
where **F**_e_ represents the elastic deformation and the rigid body rotation, and **F**_p_ represents the plastic deformation of the crystal lattice. The plastic velocity gradient **L** is defined as:(2)$$L_{p}={\dot{F}}_{p}F_{p}^{-1}=\sum_{\alpha =1}^{N}{\dot{\gamma }}^{\alpha }m^{\alpha }⊗n^{\alpha }$$
where ${\dot{\gamma }}^{\alpha }$, $m^{\alpha }$ and $n^{\alpha }$ are the plastic shear rate, slip direction vector and normal to the slip plane of the slip system *α*, respectively. A total of 12 slip systems of $\left\{111\right\}\langle 1\bar{1}0\rangle$ was considered. The shear rate of each slip system is modeled by a phenomenological power law:(3)$${\dot{\gamma }}_{s}^{\alpha }={\dot{\gamma }}_{0}{\left|\frac{\tau^{\alpha }}{\tau_{c}^{\alpha }}\right|}^{n}\mathrm{sgn}\left(\tau^{\alpha }-\chi^{\alpha }\right)$$
where ${\dot{\gamma }}_{0}$ is the reference shear rate (${\dot{\gamma }}_{0}=10^{-3}$ s), $\tau_{}^{\alpha }$ is the resolved shear stress of the slip system *α*, $\tau_{c}^{a}$ is the critical resolved shear stress (CRSS), and *n* is the strain rate exponent. The work hardening behavior due to slip-slip interaction was formulated as follows:(4)$$\tau_{c}^{\alpha }=\tau_{0}^{\alpha }+\int_{t}{\dot{\tau }}_{s\to s}^{\alpha }dt$$
(5)$${\dot{\tau }}_{s\to s}^{\alpha }=\frac{d\tau_{s\to s}^{\alpha }}{d\Gamma_{s}}\sum_{\beta =1}^{N_{s}}h^{\alpha \beta }\left|{\dot{\gamma }}_{s}^{\alpha }\right|$$
(6)$$\tau_{s\to s}^{\alpha }=\tau_{1}^{\alpha }\left\{1-\exp\left(-\left|\frac{b_{1}^{\alpha }}{\tau_{1}^{\alpha }}\right|\Gamma_{s}\right)\right\}$$
where $\tau_{0}^{\alpha }$ is the initial CRSS, $\tau_{1}^{\alpha }$ is the saturated CRSS, $b_{1}^{\alpha }$ is the initial hardening rate, $\Gamma_{s}$ is the total shear strain, and $h^{\alpha \beta }$ is the interaction coefficient of the slip systems *α* and *β*. To simulate cyclic deformation behavior, the back stress evolution was defined by the Armstrong-Frederick model as follows:(7)$${\dot{\chi }}^{\alpha }=A{\dot{\gamma }}^{\alpha }-B\left|{\dot{\gamma }}^{\alpha }\right|\chi^{\alpha }$$
where *A* and *B* are constants, each associated with a dynamic hardening and a dynamic recovery.

To calibrate the crystal plasticity parameters, strain-controlled low-cycle fatigue simulations were performed on the polycrystalline model generated in the previous section. The boundary conditions are depicted together with the model in Figure 1b. Periodic boundary conditions were applied in the x- and y-directions (RD and TD), respectively. To provide the periodic boundary conditions, equation constraints in the finite element code Abaqus were used to constrain the displacement of each node on the left end to match the displacement of the corresponding node on the right end. The top and bottom ends were similarly constrained. While maintaining these constraints, the average value of the x-displacement at the left end was fixed, and displacement (or load) was applied on the right end to simulate strain-controlled (or stress-controlled) cyclic fatigue tests. In the strain-controlled tests, the polycrystalline model was subjected to cyclic displacement in the RD with a maximum strain of 1.5%, a strain ratio of *R*_ε_ = −1, and the number of cycles at 10. The stress-strain hysteresis loop was compared with experimental data [18] and the crystal plasticity parameters were calibrated to minimize the discrepancy. The calibrated crystal plasticity parameters are shown in Table 1.

### 2.4. High-Cycle Fatigue Simulation and Crack Initiation Prediction

Stress-controlled fatigue simulations were performed to predict crack initiation in high-cycle fatigue tests. A periodic boundary condition was applied to the polycrystalline model as in the previous section, and cyclic loading was applied in the RD direction with a maximum stress of 190 to 340 MPa and a stress ratio of R = −1. The number of cycles was set to 10.

To assess the crack initiation life, an analysis based on the Tanaka-Mura model [20] was applied to the results of the crystal plasticity finite element simulations. A schematic of the crack initiation analysis is shown in Figure 2. The Tanaka-Mura model considers that a crack occurs on a slip band when the accumulated strain energy on the slip band exceeds a critical value. To specify the slip bands in the finite element model, the intersecting lines of the 2D finite element model and the slip plane were drawn at regular intervals for each slip system of each grain, as shown in Figure 2a. These lines were defined as potential crack paths. As an example, Figure 2b shows potential crack paths in a certain grain. For each potential crack path, the fatigue indicator parameter (FIP) was defined as a function of the plastic shear strain amplitude at the last cycle averaged over the path, Δ*γ*^α^, and the length of the path, *d*.
(8)$$FIP=d{\left(\frac{\Delta \gamma^{\alpha }}{2}\right)}^{2}$$

Following the Tanaka-Mura model, the crack initiation life, *N_i_*, is given by:(9)$$N_{i}=\frac{8GW_{c}}{\pi \left(1-\nu \right)d{\left(\Delta \tau -2\tau_{c}\right)}^{2}}=\frac{A_{\mathrm{TM}}}{FIP}$$
where *G* is the shear modulus, *W_c_* is the fracture surface energy per unit area, *ν* is the Poisson’s ratio. Using the FIP defined above, the crack initiation life can be predicted by the inverse FIP and a single material constant, *A*_TM_. The material constant was determined by fitting with experimental data on crack initiation life at maximum stress of 270 MPa [5]. For all potential crack paths in the polycrystalline model, *N_i_* was calculated and it was assumed that the initial crack occurs in the path with the smallest value of *N_i_*.

## 3. Results and Discussion

### 3.1. Low-Cycle Fatigue Behavior

Low-cycle fatigue simulations were repeated with varying crystal plasticity parameters until the difference in the stress-strain hysteresis loop from the experimental data was sufficiently small. The first ten hysteresis loops obtained from low-cycle fatigue experiments performed in the rolling direction are shown in Figure 3a. Basically, it is difficult to uniquely determine the solution for multiple crystal plasticity parameters based only on macroscopic stress-strain behavior. To avoid this non-uniqueness problem, among the crystal plasticity parameters, *n*, ${\dot{\gamma }}_{0}$, *A*, and *B* were fixed using literature values [9], while the other parameters *τ*_0_^α^, *τ*_1_^α^, and *b*_1_^α^ were varied. The stress-strain hysteresis loops calculated with these crystal plasticity parameters are shown in Figure 3b. The obtained stress-strain behavior agreed well with the experimental data [18], including the yield stress, the maximum stress, the loop shape, the slight cyclic hardening at the beginning, and almost saturation by 10 cycles. Other experimental investigations have also reported that the hysteresis loop reaches a saturated state after about 10 cycles at most strain levels [21,22,23].

### 3.2. High-Cycle Fatigue Behavior

Stress-controlled fatigue simulations were performed using the crystal plasticity parameters calibrated in the previous section. Based on the results of the low-cycle fatigue simulation, the number of cycles was set to 10, and the stress-strain state in the last cycle was assumed to be the same as the steady state in the high-cycle fatigue test (e.g., half the number of cycles to failure). As an example, the Mises stress distribution at maximum load in the last cycle under the conditions of a maximum stress of 340 MPa and a stress ratio of *R* = −1 is shown in Figure 4a. When the applied stress was 340 MPa, the local stress exceeded 400 MPa due to the anisotropy of the elasto-plastic behavior of the crystal grains. The macroscopic stress-strain curve is shown in Figure 4b. The highest strain appeared in the tensile loading of the first cycle, and the loops after the second cycle overlapped each other. On the compression side, almost the same loops showed from the first cycle. To examine the local deformation behavior in more detail, stress-strain curves at higher and lower stress positions were displayed in Figure 4c. At the higher stress position, the maximum tensile stress in the loading direction was 421 MPa, and at the lower stress position, the stress was 269 MPa, indicating that the stress varied about ±20% from the applied stress. Additionally, the local stress ratio showed slightly different values for each position. For example, the two elements used in Figure 4c had a stress ratio of −0.95 at the higher stress position and a stress ratio of −1.07 at the lower stress position.

### 3.3. Prediction of Fatigue Crack Initiation

The plastic shear strain amplitudes for each slip system were extracted from the high-cycle fatigue simulations in the previous section to calculate the FIP values for all potential crack paths. The cumulative plastic shear strain is shown in Figure 5 (the slip system corresponding to each contour plot is listed in Figure 2b). As can be seen from the figure, the plastic strain showed a quite heterogeneous distribution due to the plastic activities on slip systems. The FIP values of all paths were computed and the path with the highest FIP was assumed to be the crack initiation site. The predicted initial crack is indicated by an arrow in the figure. It was oriented close to 45° to the loading direction and occurred in a grain with a significant accumulation of plastic strain. This inclination of the crack to the loading direction is consistent with experimental observations [5]. Fatigue prediction studies using FIP often use volume averaging, but such methods can eliminate the intragranular heterogeneity of plastic strain. Especially for coarse grains with anisotropic geometry, the slip band size is highly dependent on the slip plane and grain shape. The concept of potential crack paths used in this paper can take into account the effect of grain shape anisotropy on crack initiation. The same crack initiation analysis was performed by changing the maximum stress in the range from 190 Mpa to 340 Mpa. The material constant *A*_TM_ in the Tanaka-Mura model was determined to be 0.66 mm by fitting based on the experimental data of the crack initiation life at an intermediate stress level, the maximum stress of 270 Mpa [5]. This value was used to predict fatigue crack initiation life at other stress levels. The predicted crack initiation life is shown in Figure 6 together with the experimental S-N curve. The predicted crack initiation life at the maximum stress of 340 Mpa was in good agreement with the experimental value of *N_i_*. Under loading conditions with no experimental crack initiation data, the predicted results of crack initiations were not contradicted with experimental failure life, *N*_f_ (the predicted crack initiation life was smaller than the *N*_f_). These results suggest that the crystal plasticity simulation proposed in this study can predict the fatigue crack initiation life in 7075 aluminum alloy. It should be noted that experimental results may be affected not only by the microstructures, but also by various factors such as specimen geometry, surface roughness, and alignment of the fatigue testing machine.

The proposed method has the advantage of accurately predicting crack initiation with only a single fitting parameter (*A*_TM_) and is useful for predicting fatigue crack initiation under various loading conditions. On the other hand, several issues remain to be addressed in order to apply the proposed method more universally to various situations. The first issue is the effect of free surfaces. In the experiments, extrusion and intrusion on an unconstrained surface can lead to fatigue crack initiation [2]. Therefore, it is necessary to include surface topology in the numerical simulations. A study that applied a crystal plasticity simulation similar to this paper to a polycrystalline model with a free surface reported that crack initiation was predicted on the surface [24]. The second issue is the crystallographic texture. It has been shown experimentally in aluminum alloys that the crystallographic texture has a significant effect on fatigue propagation and total fatigue life [25,26]. Virtual experiments based on numerical simulations also showed that the effect of the texture on fatigue crack initiation is not negligible [27]. The third issue is to model the various sources of crack initiation. It has been reported that cracks initiate from non-metallic inclusions under conditions where the stress ratio *R* = 0 or higher [3,23]. There are hard inclusions such as Al_2_Cu and soft inclusions such as Mg_2_Si, both of which can be sources of fatigue crack initiation. Another issue is the number of grains included in the polycrystalline model. Since fatigue crack initiation is a phenomenon involving scattering, fatigue crack initiation analysis should be performed on models with sufficient volume and the highest value of the FIP should be adopted. Since computational cost often limits the model size, applying an extreme value analysis [28] can be helpful. To predict total fatigue life, it is also necessary to accurately predict microstructural small fatigue crack propagation [10].

### 3.4. Physical Meaning of Tanaka-Mura Parameter

Let us consider the physical meaning of the fitted parameter, *A*_TM_. This consideration is important to make the proposed method more generally applicable to various materials. In the fatigue analysis proposed in this paper, the plastic shear strain induced by external forces and the microstructure-dependent slip length are quantified in the FIP. Hence, the *A*_TM_ is considered to involve the nature of the material. Basically, a higher *A*_TM_ means higher resistance to fatigue crack initiation. Pioneering work by Feltner et al. [29] more than 50 years ago pointed out that the stacking fault energy (SFE) affects dislocation structure changes and cyclic stress-strain responses, which are precursors to fatigue crack initiation. Then, many studies have reported that in materials with high SFE, cross slip is promoted, and subgrain structures of dislocations form under cyclic loading, leading to fatigue crack initiation along subgrain boundaries [30,31,32,33]. In their experiments, it has been difficult to separate the effect of SFE from the effect of yield stress on fatigue properties, making quantitative evaluation challenging. In the analysis performed in this paper, the effects of yield stress are separated to the FIP and the effect of SFE is considered to be included in the *A*_TM_. The *A*_TM_ values derived for various metallic materials in previous studies are listed in Table 2 together with the SFE in the literature [34,35]. Overall, the *A*_TM_ tended to increase with decreasing the SFE. This trend is roughly consistent with previous studies, as a lower SFE implies superior fatigue properties. The *A*_TM_ value of 7075 aluminum alloy obtained in this study appear to deviate from the trend. One of the reasons is that the experimental values for crack initiation life were taken from literature 30 years ago, and the spatial resolution of the observations was lower than for the other three alloys obtained more recently, resulting in an overestimation of *N_i_*. Another reason is that the 7075 aluminum alloy is precipitation-hardened, and the finely dispersed particles on the atomic order promote planar slips, as examined in detail by Gerold et al. [36]. As Mughrabi points out [37], explaining fatigue behavior with SFE is not inherently appropriate and should be compared to the ease of cross slip and the nature of dislocation slip (plane or wavy slip) because short-range order and short-range clustering in solid solution have a more significant effect on promoting planar slip than SFE. To this end, it is important to quantitatively evaluate the dislocation behavior of materials subjected to cyclic deformation, which should be discussed with numerical simulations based on discrete dislocation dynamics (DDD) [38]. In DDD, the challenge is how to formulate the dislocation mobility law, which would require MD calculations [39] and inverse analysis of dislocation motion by acoustic emission (AE) [40].

It should be noted that the *A*_TM_ also depends on the distance between the slip bands assumed in the model. Recently, high-resolution digital image correlation (HR-DIC) has made it possible to observe slip bands [41]. This technique would be useful for introducing reasonable slip-band spacing into the numerical model. Additionally, it should be kept in mind that the Tanaka-Mura model (Equation (9)) is derived from strong hypotheses, such as the irreversibility of dislocation movement, uniform shear stress on the slip band, and no work hardening. The assumption regarding slip irreversibility is inconsistent with the recent observations made by atomic force microscopy [42,43]. The observation suggested that the slip irreversibility depends on the plastic strain amplitude. One possible approach to incorporate this phenomenon is to define the fraction of the slip irreversibility as a function of plastic strain amplitude and introduce it into the Tanaka-Mura model, but there is a lack of sufficient experimental knowledge to formulate such a function. Experimental methods to measure slip irreversibility more easily are required. The DIC technique and AE method may be useful for such purposes. Previous studies have suggested that in cyclic testing of pure aluminum, the generated AE signals were associated with dislocations that move during stress reversal [44]. Further quantitative discussion of dislocation behavior during cyclic deformation will be necessary to better understand the physics of the fatigue crack initiation.

## 4. Conclusions

This work proposed a method to predict fatigue crack initiation of 7075 aluminum alloy by calculating the cyclic deformation of a polycrystalline model based on crystal plasticity simulations and substituting the results into a crack initiation criterion of the Tanaka-Mura model. The proposed method requires the crystal plasticity parameters of the material and the critical value of the Tanaka-Mura model. The following conclusions can be drawn:In both strain-controlled and stress-controlled fatigue simulations, the stress-strain hysteresis loop was saturated by 10 cycles. In the high-cycle (stress-controlled) fatigue simulations, local stress amplitudes were heterogeneously distributed in the polycrystalline model due to the anisotropic elasto-plastic deformation of crystalline grains. On the other hand, local stress ratios were almost uniform.To predict crack initiation in high-cycle fatigue experiments, potential crack paths were assumed parallel to the slip planes, and FIPs were derived on the paths and substituted into the fatigue crack initiation criterion of the Tanaka-Mura model. The predicted crack initiation life and initial crack orientation were in good agreement with the experimental results.The physical meaning of the fitting parameter (*A*_TM_) in the Tanaka-Mura model was discussed, compared with values in other alloys. Although the *A*_TM_ was weakly correlated with SFE, it was suggested that SFE is not considered the only dominant parameter on fatigue crack initiation. The importance of analyzing the dynamic dislocation behavior under cyclic loading was pointed out.

## Figures and Tables

**Figure 1 materials-16-01595-f001:**
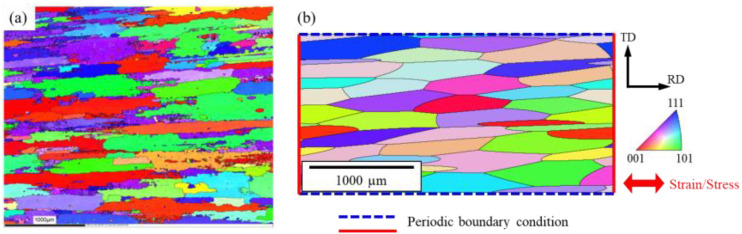
(**a**) Inverse pole figure map of 7075 aluminum alloy in the TD-RD plane (reprinted with permission from Ref. [18]), (**b**) finite element model of polycrystalline aggregate and boundary conditions.

**Figure 2 materials-16-01595-f002:**
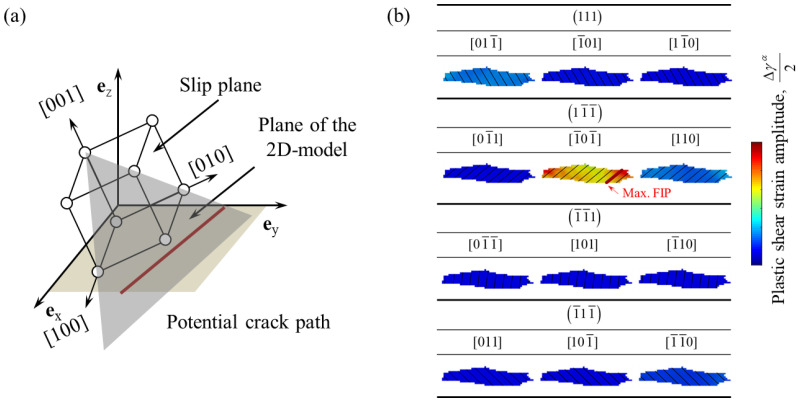
(**a**) Schematic of a potential crack path defined as the intersection of the slip plane and the 2D finite element model. (**b**) An example of the potential crack paths for a single crystalline grain. The path with the highest fatigue indicator parameter (FIP) is highlighted with a bold line.

**Figure 3 materials-16-01595-f003:**
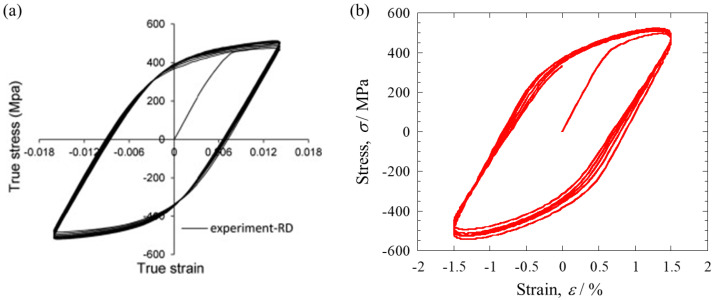
(**a**) First ten hysteresis loops obtained from low-cycle fatigue experiments performed in the rolling direction (reprinted with permission from Ref. [18]), (**b**) calibrated stress-strain hysteresis loops in low-cycle fatigue simulations.

**Figure 4 materials-16-01595-f004:**
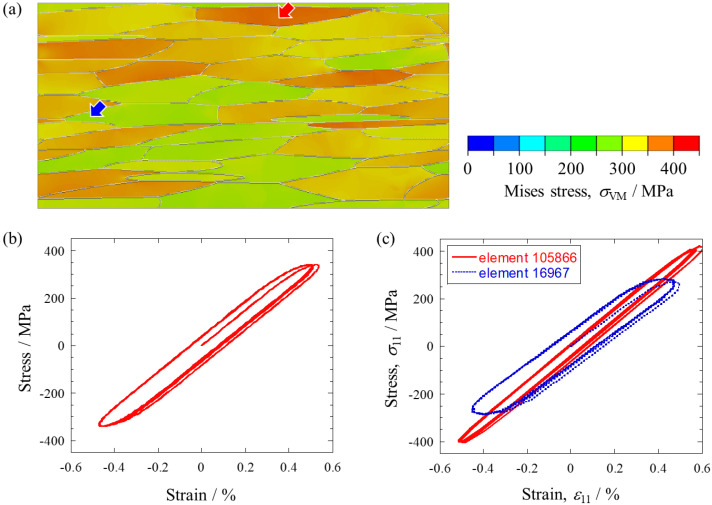
(**a**) Mises stress distribution at maximum stress applied (*σ*_max_ = 340 MPa, R = −1). (**b**) Macroscopic stress-strain curve. (**c**) Stress-strain curves at the higher and lower stress locations indicated by arrows in (**a**).

**Figure 5 materials-16-01595-f005:**
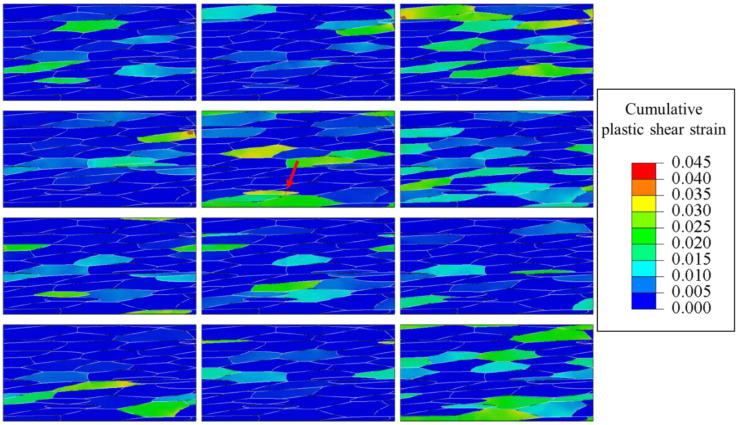
Distribution of cumulative plastic shear strain for 12 slip systems in the last calculation step (*σ*_max_ = 340 MPa, R = −1). The predicted crack initiation site is indicated by an arrow and a red solid line.

**Figure 6 materials-16-01595-f006:**
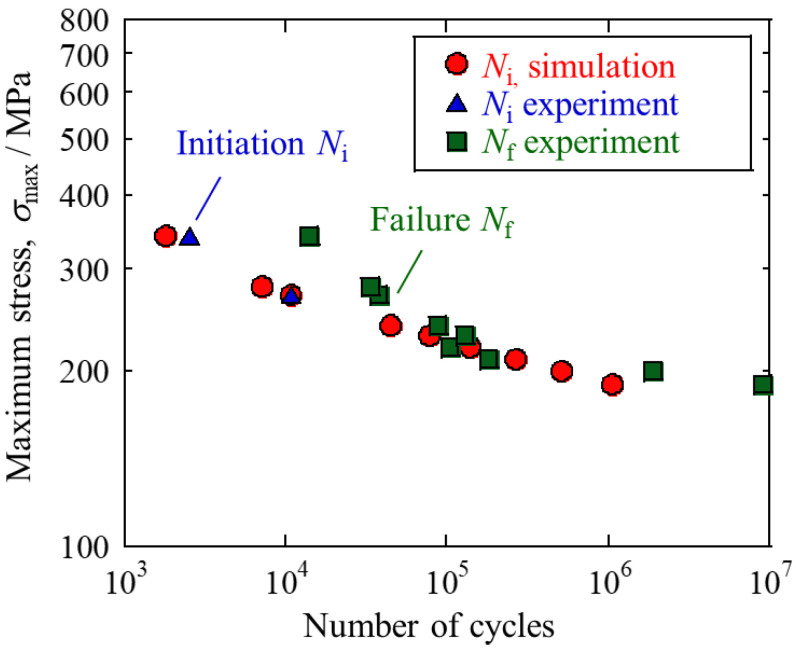
Predicted fatigue crack initiation life and experimental fatigue crack initiation and failure life taken from the literature [5].

**Table 1 materials-16-01595-t001:** Elastic constants and crystal plasticity parameters for aluminum alloy 7075.

Property	C_11_	C_12_	C_44_	${\dot{\gamma }}_{0}$ □	*n*	$\tau_{0}^{\alpha }$ □	$\tau_{1}^{\alpha }$ □	□ $b_{1}^{\alpha }$	*A*	*B*
Unit	GPa	GPa	GPa	s^−1^	-	MPa	MPa	MPa	MPa	-
Value	106.75	60.41	28.34	0.001	7.14	150	160	200	8100	80

**Table 2 materials-16-01595-t002:** Tanaka-Mura parameter and stacking fault energy (SFE). The SFE is the literature value of the base metal of each material.

Material	Tanaka-Mura Parameter/mm	Stacking Fault Energy /mJ/mm^2^	Crystal Structure
Ti-6Al-4V alloy	0.0014 [15]	310–82 [33]	HCP
Fe	0.14 [14]	180 [34]	BCC
7075 Al alloy	0.66	166 [34]	FCC
Mg-Al-Ca-Mn alloy	0.245 [16]	125 [34]	HCP

## Data Availability

The raw/processed data are available from the corresponding author upon reasonable request.

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
