# Peer review of "Prediction of Fatigue Crack Initiation of 7075 Aluminum Alloy by Crystal Plasticity Simulation"

_materials, 2023, doi:10.3390/ma16041595_

Round 1

Reviewer 1 Report

This paper proposed a method for predicting fatigue crack initiation of 7075 aluminum alloy by crystal plasticity finite element analysis considering microstructures. The local plastic strain calculated by CPFEM was introduced into the crack initiation criterion (Tanaka and Mura, 1981) to predict fatigue crack initiation life. There are some problems in this paper and it should be rejected. The details as follow:

1.      The logic of the paper is confused. First, the introduction is too simple. Second, many parts of this paper are not in the location that they should be. Such as part 3.1 is model parameters calibration, it belongs to part 2.3, experiment curve should be given in Figure 3, and the issues remain to be addressed in part 3.3 are too much.

2.      The experiment results should be given in part 3.3 so that comparative analysis can be carried out. The authors present the concept of potential crack paths, but it is not emphasized in the model results. The importance of this for life prediction should be highlighted.

3.      RVE is representative volume element, the authors should give the SEM and EBSD images of as-received Al alloy material, in order to show the RVE is a reasonable model to experiment material microstructure.

4.      The crystal plasticity equations are incomplete, the authors should give the equations of total deformation gradient F,plastic velocity gradient Lp and others.

Reviewer 2 Report

The authors proposed a method for predicting fatigue crack initiation of 7075 aluminum 10 alloy by crystal plasticity finite element analysis considering microstructures, which sounds interesting for reader. This manuscript can be accepted by Materials, however, several points should be considered before acceptance. 

1. The authors claiemd that the simulation has been verified by the experimental data for literature. However, I do not think the experimental data are enough to validate the prediction. I suggest authors should some experiments to verify it and convince the readers.

2.  The formate of references should be unified.

Reviewer 3 Report

In the article Prediction of Fatigue Crack Initiation of 7075 Aluminum Alloy by Crystal Plasticity Simulation, a new method for analyzing and predicting the occurrence of a fatigue crack in 7075 aluminum alloy is proposed. literature data using simulation. In general, this article is quite promising and interesting not only from a practical point of view, but also from an experimental one, as it offers a new method for predicting the formation of fatigue cracks in materials, which complements previously known methods. According to the reviewer, this article can be accepted for publication after the authors answer a number of questions that have arisen during its reading and analysis.

1. First, the authors should explain why this 7075 aluminum alloy was chosen as the main objects of study. In the introduction, the authors say that this alloy is quite promising in the aerospace industry, but this should be briefly mentioned in the abstract.

2. The authors should give more details about the experimental work carried out on cyclic tests.

3. In conclusion, a number of parameters should be given that complement the standard Tanaka-Mura model.

4. Cyclic tests presented in Figure 6 show a rather sharp decline in stability after 1e5 test cycles, the authors should give a more detailed explanation of this.

5. The data presented in Table 1 require additional presentation of the measurement error and standard deviation values in order to understand that these values were not obtained randomly.

Reviewer 4 Report

In this work, a constitutive law based on crystal plasticity theory and the Tanaka-Mura model were used in the finite element analysis. But there are still some problems in the manuscript. The manuscript needs the necessary revisions as stated below:

1) Figure 2(b) shows potential crack paths in a certain grain, these callouts can be enlarged for better understanding。

2) The marks in the figure 4(a) and figure 5 can give a local enlarged view. 

3) Section 5 is actually Section 4, please check for text errors.

4) Grammar corrections need to be done to improve the quality of the manuscript. 

5) For constitutive parameters, why the value of reference shear rate is 10-3s? Please provide the reference. (The references of other parameters are given in the work.)

6) Periodic boundary conditions are applied in the simulation. How is the stress and strain extracted in Figure 3 and Figure 4(b)? In addition, periodic boundary conditions should be appropriately described.

7) As described in Section 3.3 “The second issue is the crystallographic texture.”. The method of constructing the polycrystalline model in this paper is random, how to consider this randomness?

Round 2

Reviewer 1 Report

This paper proposed a method for predicting fatigue crack initiation of 7075 aluminum alloy by crystal plasticity finite element analysis considering microstructures. The local plastic strain calculated by CPFEM was introduced into the crack initiation criterion to predict fatigue crack initiation life. There are some problems in this paper and it should be published after major revision. The details as follow:

 1.     The introduction is still too simple. Introduction is a part that used to show how the authors' understanding of the research field. The authors should divide the introduction into at least three parts: 7075 aluminum alloy, prediction of fatigue crack initiation and crystal plasticity model. This shows that the author has conducted enough investigation on the research content of this paper.

2.     The number of elements is a important factor to measure the effectiveness of finite element model, authors should give how much the elements does the model have.

Reviewer 2 Report

Authors have addressed the raised concerns and the efforts improved the quality of work at very significant level. Therefore, the reviewer is agreed to publish this work in Materials.

Author Response

The authors would like to thank reviewers again for their enlightened and useful comments to help us improve the paper.

Reviewer 3 Report

The authors answered all the questions posed, the article can be accepted for publication.

Author Response

(The authors gave the same response as above.)

Round 3

Reviewer 1 Report

No other comments.